# Physiological Variability during Prehospital Stroke Care: Which Monitoring and Interventions Are Used?

**DOI:** 10.3390/healthcare12080835

**Published:** 2024-04-15

**Authors:** Abdulaziz Alshehri, Jonathan Ince, Ronney B. Panerai, Pip Divall, Thompson G. Robinson, Jatinder S. Minhas

**Affiliations:** 1Cerebral Haemodynamics in Ageing and Stroke Medicine (CHiASM) Research Group, Department of Cardiovascular Sciences, University of Leicester, Leicester LE1 7RH, UK; aaha7@le.ac.uk (A.A.); rp9@le.ac.uk (R.B.P.); tgr2@le.ac.uk (T.G.R.); 2College of Applied Medical Sciences, University of Najran, Najran P.O. Box 1988, Saudi Arabia; 3NIHR Leicester Biomedical Research Centre, British Heart Foundation Cardiovascular Research Centre, Glenfield Hospital, Leicester LE3 9QP, UK; 4University Hospitals of Leicester NHS Trust, Leicester LE1 5WW, UK; pip.divall@uhl-tr.nhs.uk

**Keywords:** prehospital, stroke, physiological variables, end-tidal CO_2_, blood pressure, heart rate, mortality, long-term outcomes

## Abstract

Prehospital care is a fundamental component of stroke care that predominantly focuses on shortening the time between diagnosis and reaching definitive stroke management. With growing evidence of the physiological parameters affecting long-term patient outcomes, prehospital clinicians need to consider the balance between rapid transfer and increased physiological-parameter monitoring and intervention. This systematic review explores the existing literature on prehospital physiological monitoring and intervention to modify these parameters in stroke patients. The systematic review was registered on PROSPERO (CRD42022308991) and conducted across four databases with citation cascading. Based on the identified inclusion and exclusion criteria, 19 studies were retained for this review. The studies were classified into two themes: physiological-monitoring intervention and pharmacological-therapy intervention. A total of 14 included studies explored prehospital physiological monitoring. Elevated blood pressure was associated with increased hematoma volume in intracerebral hemorrhage and, in some reports, with increased rates of early neurological deterioration and prehospital neurological deterioration. A reduction in prehospital heart rate variability was associated with unfavorable clinical outcomes. Further, five of the included records investigated the delivery of pharmacological therapy in the prehospital environment for patients presenting with acute stroke. BP-lowering interventions were successfully demonstrated through three trials; however, evidence of their benefit to clinical outcomes is limited. Two studies investigating the use of oxygen and magnesium sulfate as neuroprotective agents did not demonstrate an improvement in patient’s outcomes. This systematic review highlights the absence of continuous physiological parameter monitoring, investigates fundamental physiological parameters, and provides recommendations for future work, with the aim of improving stroke patient outcomes.

## 1. Introduction

Stroke care has rapidly advanced over the last decade, predominantly focusing on the delivery of mechanical thrombectomy (MT) in acute ischemic stroke (AIS) and the reduction in the time between diagnosis and definitive care (such as through the use of prehospital stroke scales, mobile stroke units [MSUs], and standardized care pathways) [1,2,3,4]. Despite this, stroke is still a leading cause of mortality and morbidity, responsible for 11.6% of global annual deaths [5]. Furthermore, its estimated incidence is expected to continue to rise, with estimates of a 27% increase between 2017 and 2047 in the European Union [6]. As a result, interventions focused on improving patient outcomes in stroke is highly topical and will likely have ever increasing importance in healthcare guidelines.

The physiological parameters monitored in acute stroke are increasingly being considered with respect to how they can affect patient outcomes. As has been documented in the literature, altered physiological states (such as hypoxia, hypocapnia, hyperglycemia, and hypotension) have been shown to be associated with deteriorating neurological condition and increased [7,8,9,10,11]. As a result, the question has been raised of whether the introduction of advanced prehospital physiological monitoring and intervention would be able to improve the outcomes of patients. Blood pressure (BP) is the most commonly explored parameter in the recent literature, with the Paramedic Acute Stroke Treatment Assessment (PASTA) and RIGHT-2 randomized control trials having looked at altering BP in patients with acute stroke [12,13]. Despite this, there appears to be little consensus on the optimal ways of monitoring and managing physiological parameters prehospitally in patients with acute stroke. Physiological differences were found between stroke and stroke mimics in the prehospital setting [14], including blood glucose levels, systolic and diastolic blood pressure, body temperature, oxygen saturation, and heart rate (HR). Although intermittent measures showed changes, continuous physiological parameter measurements could provide even better information and prove more predictive than isolated measures [15].

Previous works have emphasized the feasibility and potential benefits of continuous monitoring in acute stroke, particularly for BP, end-tidal carbon dioxide levels (EtCO_2_), HR, and glycemic control [16,17]. Two studies compared the outcomes of MSUs with those of conventional prehospital stroke care [18,19]. The MSU is typically a specialized ambulance equipped with a portable CT scanner, a point of care laboratory, and stroke medication, as well as telemedicine interactions for expert consultation between the ambulance and the hospital [20]. It is staffed by a paramedic, emergency medicine and neurology specialists, and a radiology technician [19]. It was demonstrated that MSU management had a higher percentage of patients with (mRS) of 0–2 at 90 days compared to conventional care, in addition to a reduction in the mean time to CT and intervention [18,21]. With the growing focus on prompt management following stroke recognition, and 80% of stroke patients arriving to UK hospitals doing so by ambulance [22], we hypothesize that physiological parameter monitoring and intervention in the prehospital environment may improve patient care. This systematic review aims to investigate the current use of prehospital physiological monitoring and interventions to modify these parameters. Parameters of particular interest include BP, HR, respiratory rate (RR), EtCO_2_, and oxygen saturations.

## 2. Materials and Methods

### 2.1. Study Identification

This systematic review was conducted and reported following the guidelines for the Preferred Reporting Items for Systematic Reviews and Meta-analysis (PRISMA) [23]. The protocol for this review was registered on PROSPERO on 11 March 2022 (CRD42022308991).

### 2.2. Search Strategy

A search strategy was developed with a clinical librarian (PD) to identify relevant studies across four databases (MEDLINE, EMBASE, CINAHL and CENTRAL) from their inception to August 2022, accommodating various Medical Subject Heading (MeSH) terms or subject headings available on each database. The keywords included were “stroke”, “prehospital”, “haemodynamic variables”, “physiological variables”, “end-tidal CO_2_”, “blood pressure”, “heart rate”, “mortality”, “long-term outcomes” (Appendix A). Furthermore, a grey literature search was performed using the reference lists and citation indices of the included studies, in addition to the use of the Google Scholar search engine to identify further relevant articles. Rayyan QCRI web-tool (Qatar Computing Research Institute) and Endnote X9 were used to eliminate duplicate articles and to review retrieved records [24]. Two reviewers (AA and JI) independently screened studies according to the selection criteria disagreements were addressed by engaging in discussions with a third reviewer (JM). 

### 2.3. Inclusion and Exclusion Criteria

Studies were included if they met the following criteria: (1) all study types were included; (2) English full-text studies that included patients with suspected acute stroke receiving prehospital care; and (3) studies that involved adult patients (≥18 years). 

Exclusion criteria were as follows: (1) non-human studies; (2) studies that included patients presenting with stroke who have not received prehospital care; (3) non-English-language texts; and (4) full text not available.

In cases where the same data from a single study were published in multiple records, the publication with the greatest sample size was selected.

### 2.4. Data Extraction

Two independent authors (AA and JI) screened the studies’ titles and abstracts to determine whether they met the inclusion criteria and extracted all the relevant data as follows: (1) study characteristics (publication year, first author, study type, country, sample size and stroke type); (2) study main objective; (3) interventions that were delivered to optimize the physiological parameters; and (4) outcomes (main conclusions and results).

### 2.5. Risk of Bias Assessment

The methodological quality of the included studies was assessed by two reviewers (AA and JI) independently using the Newcastle-Ottawa scale (NOS) [25]. The NOS was selected as it best suited the study types included in the review. Discrepancies in scoring were discussed and resolved by consensus. This scoring system examines the quality of an article based on three main aspects: selection of study (4 points), comparability of groups (2 points), and evaluation of outcomes (3 points). A score of ≥7 points indicated a high-quality study. Appendix A presents the risk of bias assessment of the included studies.

### 2.6. Data Synthesis

Studies were organized into those looking at physiological monitoring and pharmacological interventions. Further sub-divisions were based on the studied physiological parameter (such as HR, BP, oxygen saturation, and EtCO_2_). Stroke type (such as AIS or ICH) and clinical outcome (typically assessed using the modified Rankin scale (mRS)) were also considered.

The heterogeneity assumption was planned to be checked by the χ^2^-based Q test. And I^2^ value of >50% or a *p*-value of <0.05 for the Q2 statistic was intended to indicate significant heterogeneity. However, meta-analysis of the included data was later considered not to be possible due to the heterogeneity in terms of population and different outcome measures that appeared after extracting and collecting the data.

## 3. Results

### 3.1. Study Selection

A total of 741 articles met the search criteria and were evaluated, of which 35 duplicates were eliminated, leaving 706 records for screening (Figure 1). Following title and abstract screening, 640 records were discarded due to irrelevance. Full-text review of 66 retrieved records was performed. Of these, 47 records were excluded for the following reasons: one full article not in English, some studies were on an animal subject (*n* = 41), one study was not conducted in a prehospital setting, only the study protocol but not the main studies were published (*n* = 2), and two studies did not measure any physiological variables. In total, 19 records were included in this review. 

### 3.2. Main Findings

Studies were classified into two themes: physiological monitoring intervention and pharmacological therapy intervention.

#### 3.2.1. Physiological Monitoring Intervention 

Blood pressure and patient outcomes

A total of 14 included studies explored prehospital physiological monitoring, with the investigation of the relationship between prehospital BP and patient clinical or radiological outcomes (defined in a variety of measures) being the most commonly investigated [26,27,28,29,30,31,32].

In 2015, Fan et al. investigated the relationship between prehospital BP and END (defined in this study as ≥2-point decrease in the Glasgow Coma Scale (GCS) within 24 h of emergency department (ED) arrival) in 536 patients with spontaneous ICH [26]. Significant associations between increasing on-scene systolic BP (SBP), diastolic BP (DBP), and mean arterial pressure (MAP) with END were reported following covariate-adjusted analysis [26]. The odds ratios and 95% confidence intervals (CI) were calculated per 10 mmHg increase of each parameter, with reported associations of: SBP (OR: 1.126, 95% CI: 1.015–1.265), DBP (OR: 1.146, 95% CI: 1.019–1.303), and MAP (OR: 1.225, 95% CI: 1.057–1.443) [26]. Furthermore, it was noted that prehospital BP measurements yielded higher regression model fitness than ED or neurocritical care unit BP measures [26]. 

These findings were further supported by Tsou et al., who identified a similar association in patients with spontaneous ICH, whereby patients with an increase in prehospital SBP of more than 15 mmHg had an increased risk of END [27]. However, it must be noted that END had a different definition, instead being defined as a ≥2-point decrease in GCS within 6 h of ED arrival [27].

Building on this previous work, Larsen et al. also investigated the relationship of prehospital BP on clinical and radiological outcomes for patients with spontaneous ICH [28]. Unlike findings by Fan et al. and Tsou et al., there were no significant differences in prehospital BP measures between patients experiencing END (defined at 24 h from ED arrival) and those not [28]. However, across this sample of 426 patients, there were linear associations between prehospital DBP and MAP with hematoma expansion (OR: 1.10, 95% CI: 1.00–1.21 and OR: 1.09, 95% CI: 1.00–1.18, respectively) [28]. Prehospital MAP was also linearly associated with in-hospital death (OR: 1.08, 95% CI: 1.01–1.15) [28]. Finally, there was an association reported between prehospital SBP and predicted probability of in-hospital death, best explained by a non-linear relationship (*p* = 0.017), with the nadir of the U at 160 mmHg [28].

In addition to investigation of END, the literature has also investigated whether early BP monitoring could predict prehospital neurological deterioration (PND), defined as negative changes to the patient’s GCS during the prehospital period [29,30]. Whilst similar to END, this earlier deterioration has been associated with reduced functional independence and increased mortality [33], prompting further investigation. 

Atsumi et al. confirmed the importance of investigating PND, as this early deterioration in a population with spontaneous ICH was associated with an unfavorable clinical outcome (defined as modified Rankin Scale (mRS) of >2, *p* = 0.20) and needing emergency surgery or death within 24 h (*p* < 0.0001) [29]. Within this population, significantly raised SBP was reported in patients with PND (*p* = 0.0079), in addition to being associated with hematoma enlargement between initial and repeat CTs (*p* = 0.005) and needing emergency surgery or death within 24 h of ED arrival (*p* = 0.0002) [29].

However, the association between prehospital BP and PND in stroke is not entirely clear. In a large retrospective study by Slavin et al. of 1092 acute stroke patients (866 AIS, 175 ICH, and 51 SAH), there was no significant difference in prehospital SBP or DBP measures between PND and non-PND groups [30]. This absence of relationship was further extended when assessing the ICH subgroup, with no reported significant association [30]. There were however differences in PND rates between stroke subtype, with ICH/SAH (combined for multivariable analysis) having a significantly higher association with PND (OR: 3.13, 95% CI: 2.03–4.86) when compared to AIS [30]. Also, of note in this study, prehospital glucose level was significantly associated with PND among the 175 patients with ICH (OR: 0.99, *p* = 0.03) [30].

Alongside the radiological outcomes by Larsen et al. mentioned above, Hatcher et al. and Rodriguez-Luna et al. have also investigated whether prehospital SBP is associated with ICH volume [31,32]. In 2017, Hatcher et al. demonstrated across a sample of 180 patients with spontaneous ICH and documented prehospital SBP, 96% had elevated SBP, with a multivariable regression model showing an association between elevated SBP (>140 mmHg) and larger hematoma volume (OR: 3.85, 95% CI: 1.02–4.60) [31]. These findings were reinforced in 2018 by Rodriguez-Luna et al. who found that across a sample of 219 patients with ICH (of which 126 were considered hyperacute as they had imaging within 6 h of symptom onset), patients with higher ICH volumes had significantly higher prehospital SBP values (OR: 1.01 for a 1-U increase in SBP, 95% CI: 1.01–1.02, *p* = 0.018) [32]. 

The effect of variations in repeated BP measurements during prehospital transportation has also been investigated [34]. A prospective study of 91 patients with AIS eligible for thrombectomy identified that patients who had lower SBP at all five time points during prehospital transportation had significantly improved clinical outcomes (defined as whether the patient had neurological worsening, measured by NIHSS) [34]. Furthermore, a linear regression comparing change of MAP between first and last recordings during transportation found a correlation between a reduction in MAP and higher NIHSS (*p* = 0.03, β = 0.27) [34].

Finally, rather than the use of isolated BP measurements, BP variability (BPV) has also been considered. BPV is a measure of fluctuations in BP which can be characterized as very short-term (beat-to-beat), short-term (over 24 h), mid-term BPV (day-to-day) and long-term BPV (daily or monthly) [35]. Kench et al. studied the association between BPV and functional outcomes (defined as mRS ≥ 2 at 3 months) in AIS patients who had hemorrhagic transformation following thrombolysis [36]. A univariate analysis indicated that higher systolic BPV was associated with symptomatic ICH and mortality [36]. Additionally, higher systolic BPV was independently associated with worse functional outcomes (OR = 1.68, 95% CI: 1.05–2.69, *p* < 0.05) [36].

In considering our previously stated need to reduce time between diagnosis and definitive treatment, relevant literature exists. Kench et al. investigated reducing delays on scene and providing rapid transport to hospital or a hyperacute stroke unit if available based on The Joint Royal Colleges Ambulance Liaison Committee (JRCALC) guidelines recommendations [36]. In addition, Tsou and slavin found that several BP measurements in a prehospital setting and transportation time itself were not significantly associated with END and PND [27,30].

Other uses of blood pressure measurements

In addition to the above retrieved records investigating the relationship between prehospital BP and patient outcomes, this search also identified studies describing the use of BP measurements in different domains.

Firstly, Gioia et al. investigated whether prehospital BP patterns could be used to distinguish between stroke and stroke mimics [37]. They identified that prehospital SBP was significantly higher in acute stroke (155.6 mmHg, 95% CI: 153.4–157.9) compared to stroke mimics (146.1 mmHg, 95% CI: 142.5–148.6), *p* < 0.001, across a sample of 960 patients transported for suspected stroke [37]. Furthermore, the study demonstrated that SBP was higher in ICH compared to AIS and transient ischaemic attack (TIA), but also that BP measurements by prehospital providers and ED were similar [37].

Next, in 2013, Asaithambi et al. performed a retrospective analysis of 40 patients presenting with AIS, to ascertain adherence to standards when transferring patients receiving intravenous thrombolysis from primary stroke centers (PSC) to comprehensive stroke centers (CSC) [38]. BP was a key recorded parameter, with measurements being taken at two intervals in transportation (10 and 20 min). Mean SBP during transportation (143.6 ± 26.9 mmHg) was similar to the measurements take upon arrival at CSCs (146.6 ± 23.3 mmHg) [38], supporting the use of prehospital BP measurements as part of the patient’s longitudinal care record.

Gioia et al. reported relatively stable prehospital BP among stroke patients who had at least two sets of measurements, which could be attributed to the brief time spent in transportation, which was 45.0 min [37]. Asaithambi et al. showed mean prehospital transportation time 37.7 ± 20.2 min which did not differ between adherence and no nonadherence to guidelines [38].

Other physiological parameters

The remaining included records observed different prehospital physiological measurements, including EtCO_2_, HRV, and SpO_2_. 

In 2020, a retrospective study was conducted to compare the effectiveness of rapid sequence intubation (RSI) in treating prehospital stroke patients (43,831) compared to patients with traumatic brain injury (TBI) (63,297) [39]. This was to determine whether evidence from TBI-related RSI studies could be applied to stroke-related RSI. The study reported multiple prehospital physiological parameters, including BP, HR, RR, EtCO_2_ and SpO_2_. Feasibility of measurements in the prehospital environment were demonstrated, although due to the purpose of this study, the effect of these parameters on stroke were not specified. 

Additionally, a single observational study of 40 patients (with a subgroup of ten stroke patients) assessed the association between prehospital HRV and unfavorable outcome, defined as either need for admission to the intensive care unit (ICU), death during hospital admission, or need for prolonged hospital stay (>30 days) [40]. HRV was measured using four different methods, including time domain parameters, frequency domain parameters, non-linear analysis, and time-frequency analysis [40]. It was identified that time domain, frequency domain, and non-linear analysis measurements of HRV were correlated with increased risk of unfavorable outcome across the whole patient cohort (*p* < 0.05) [40]. Furthermore, a univariate analysis revealed that the frequency and time-frequency domains were positively correlated with SpO_2_ measurements (*p* = 0.376; *p* = 0.017 and ρ = 0.372; *p* = 0.018 respectively) and inversely correlated with glycemic measurements (ρ = −0.424; *p* = 0.039 and ρ = −0.421; *p* = 0.040, respectively). Ambulance transportation time was recorded as 6 min (IQR, 5–8 min), and HRV measurements were noted as lasting 5 min during transportation. [40].

Finally, a retrospective described the general mechanical ventilation (MV) settings used by emergency care providers among patients with spontaneous ICH and suspected high ICP who had inter-hospital ED transfer, and compared the hemodynamic parameters and hospital mortality between MV patients and non-MV patients in the ED [41]. In this study the median value multiple physiological parameters that were measured include partial pressure of CO_2_ 42 (35–49), respiratory rate 15 (14–18), SBP at two intervals. The study revealed the MV group was associated with a significantly higher in-hospital mortality rate (30% MV, 13% non-MV OR 2.88, 95% CI 1.6–5.19, *p* < 0.001) and higher median admission SBP in the ED (181 mmHg MV, 163 mmHg non-MV, *p* = 0.005). Results of this study were limited to those presented in the abstract.

#### 3.2.2. Pharmacological Therapy Intervention

Seven included records investigated the delivery of pharmacological therapy in the prehospital environment for patients presenting with acute stroke. Primary targets of therapy included BP reduction, neuroprotection [13,42,43,44,45]. 

Three randomized control trials (RCTs) evaluated the effectiveness of prehospital BP lowering interventions [13,42,43]. In 2013, Ankolekar et al. reported the rapid intervention with glyceryl trinitrate in hypertensive stroke trial (RIGHT), where patients with suspected acute stroke were randomly assigned to either a glyceryl trinitrate (GTN) patch or placebo [42]. Of the 41 enrolled patients, 25 were assigned to the GTN group [42]. Those assigned to the GTN group had a significantly lower SBP than those in the control group, with a significant difference of 21 mmHg at 15 min post-randomization, and 18 mmHg at 2 h [42]. There were also reported significant differences in clinical outcome for the GTN group, with 90-day mRS being shifted by 1 point across all participants (*p* = 0.040), and by 2 points when focusing on participants confirmed to only have acute stroke (*p* = 0.017) [42]. No difference in mortality or severe adverse events were found between the two arms of the study [42]. It must however be noted that due to the small sample size, the study was not powered to assess functional outcome [42]. 

Following the work of the RIGHT study, the RIGHT-2 study was reported in 2019 [13]. In this multicenter, phase 3 RCT, patients were randomly assigned in a 1:1 order to receive either transdermal GTN for 4 days or a sham dressing [13]. In this study, 1149 participants were recruited, with 852 participants confirmed to have acute stroke [13]. The GTN group was found to have significantly lower SBP (5.8 mmHg, *p* < 0.0001) and DBP (2.6 mmHg, *p* = 0.0026) when compared to the sham group, but there were no reported differences in HR [13]. Unlike the findings of the RIGHT study, there was no significant difference in functional outcome (measured using 90-day mRS) between the two groups [13]. In fact, Bath et al. acknowledged a possible tendency towards harm when using GTN for patients with ICH, very early stroke (<1 h) and severe stroke (GCS < 12, NIHSS > 12) [13].

Lastly, the paramedic initiated lisinopril for acute stroke treatment (PIL-FAST) RCT investigated the feasibility of starting definitive prehospital BP-lowering therapy [43]. Recruited patients were randomly assigned to either lisinopril or placebo [43]. Whilst the study showed a BP reduction of 14 mmHg in the lisinopril group at 24 h, the study was limited by only 14 recruited participants, of which only 4 completed the full 7 days of study medication [43]. Despite this, feasibility of prehospital physiological monitoring and treatment was demonstrated in acute stroke. The study demonstrated a lower median duration of transport time from the scene to arrival at the ED (25 min) among the intervention group compared to the placebo group (38 min). 

The role of neuroprotection in the prehospital environment has also been considered. The Field Administration of Stroke Therapy–Magnesium (FAST-MAG) trial was conducted to determine the benefit of administering magnesium sulfate as a neuroprotective agent for stroke patients in the prehospital setting within 2 h after the onset of symptoms [44]. Among the 1700 stroke participants (857 in the magnesium arm and 843 in the placebo arm), there was no significant difference in 90-day mortality (132 (15.4%) and 131 (15.5%), respectively) or mean 90-day mRS (2.7 in each group, *p* = 1.00). The duration of transport time from the scene to arrival at the ED was measured and no substantial difference was shown between the magnesium group (with a mean of 32 min) and the placebo group (with a mean of 33 min). The trial concluded that the initiation of magnesium sulfate was safe in the prehospital setting but did not demonstrate any improvement in disability scores [44]. 

Next, in 2020, Dylla et al. considered the role of prehospital oxygenation in acute stroke [45]. It was hypothesized that hyperoxia through supplementary oxygen may increase cerebral oxygenation to penumbral tissue in acute stroke, so this retrospective analysis study of 1352 stroke patients was conducted [45]. Patients were categorized into three groups: “hypoxia” (*n* = 144) who received oxygen supplementation for hypoxia, “normoxia” (*n* = 848) who did not receive oxygen, and “hyperoxia” (*n* = 360) who received oxygen despite being “normoxic”. The study demonstrated that hyperoxic participants, when compared to normoxic subjects, had significantly lower SBP (142.9 vs. 148.9 mmHg, *p* = 0.007), DBP (78.9 vs. 85.2 mmHg, *p* < 0.001), and MAP (100.3 vs. 105.3 mmHg, *p* < 0.001). The study found no significant difference in mRS at discharge in all groups [45].

## 4. Discussion

This systematic review explored the existing literature on prehospital physiological monitoring and pharmacological therapy intervention among patients with acute stroke. This review demonstrated that a variety of physiological monitoring techniques have been demonstrated in the prehospital environment, with some associations drawn between abnormal prehospital measurements and negative patient outcomes. The most commonly investigated physiological parameter was BP, with hypertension associated with increased hematoma volume in ICH and some reports of increased rates of END and PND with increased BP. However, the associations with END and PND were less conclusive, with inconsistent results demonstrated within the literature, which may be attributable to the study heterogeneity and varying outcome definitions. The literature also demonstrated that a reduction in prehospital HRV was associated with unfavorable clinical outcomes. Despite this, it was noted that there is an absence of studies primarily focusing on other key parameters, including EtCO_2_, glycemic control, and SpO_2_.

In addition to physiological monitoring, this review highlighted those pharmacological therapies that have been successfully implemented in a prehospital environment. BP lowering interventions (using GTN and lisinopril) were successfully demonstrated through the RIGHT, RIGHT-2, and PIL-FAST studies, however evidence of their overall benefit to clinical outcomes is limited. Studies investigating the use of oxygen and magnesium sulfate as neuroprotective agents did not demonstrate improved functional outcomes for patients with acute stroke. 

The prehospital balance between continuous monitoring and intervention against minimizing door-to-intervention time is crucial to optimize patient outcomes while minimizing definitive care delays. By having robust technology and efficient systems in place, it is certainly feasible to achieve both prehospital stabilization of patients and rapid transportation to appropriate treatment facilities. This review has also illustrated the prehospital balance that weighs prehospital monitoring and intervention against reducing the door-to-intervention time (Figure 2). 

Physiological monitoring interventions:

Based on the literature that investigated the association between prehospital BP and clinical outcomes among patients with spontaneous ICH patients, two studies included in this systematic review revealed a positive significant correlation between SBP, DBP, MAP, and END [26,27]. However, result contrary to these finding must be acknowledged. Larsen et al. [28] found no substantial differences in prehospital BP recordings between patients with and without END. These unexpected findings might be attributed to differences in study design and the variability in defining the outcomes. 

Furthermore, the literature has also examined the potential for early BP monitoring to predict prehospital neurological deterioration (PND). In this review, there was conflicting evidence regarding whether BP parameters are associated with PND among spontaneous ICH patients [29,30]. With this in mind, further research is needed to investigate this relationship. 

The correlation between prehospital BP and radiological outcomes was described by multiple studies within this review. Larsen et al. identified an association between DBP, MAP, and haematoma expansion [28], whilst Rodriguez-Luna et al. and Hatcher et al. revealed the significant correlation between ICH volume and elevated SBP [31,32]. These findings are similar to hospital-based studies which found that hematoma expansion was more prevalent in patients with SBP > 140 mmHg than those with a lower SBP [46]. 

Whilst absolute measurements of BP were predominantly used in the retrieved records of this review, we question whether BPV may offer greater insight into the associations between patient BP and clinical/radiological outcomes. A recent study investigated the correlation between BPV and associated outcomes among ICH patients enrolled in the FAST-MAG trial [47]. It was found that greater BPV during the hyperacute period (15 min to 5 h after onset) was associated with poorer functional outcomes at 30 days [47]. Additionally, in a systematic review of 18 studies examining the potential prognostic value of BPV in ICH and AIS patients [48], Manning et al. observed a correlation between higher systolic BPV and worse long-term functional outcomes in participants with acute stroke [48]. Considering the association between high BPV and poor outcomes that has been demonstrated during the ultra-acute phase among ICH patients, and the correlation between elevated systolic BPV and poor long-term outcomes among stroke subtypes, stroke patients may benefit from sustained and continuous BP control rather than a focus on absolute BP values. Future research is required to explore the potential of using BPV as a modifiable therapeutic target during the vulnerable period, particularly in the prehospital setting.

HRV is another under investigated parameter within the prehospital setting. In addition to the retrieved work by Yperzeele et al., which found an association between several HRV parameters and unfavorable clinical outcomes [40], a systematic review of in-hospital HRV measurements found that HRV can serve as a predictor for stroke outcomes, including stroke severity, mortality, and functional outcomes [49]. The feasibility of prehospital use of HRV has not only been demonstrated by Yperzeele et al., but by two studies that investigated the effect of HRV on patient outcomes during prehospital trauma care [50,51]. Cooke et al. found that a reduction in the HRV indices were associated with a higher risk of mortality in severe trauma patients [50], whilst King et al. demonstrated that HRV parameters provide a significant contribution as a part of a prehospital trauma triage tool. [51]. HRV has, therefore, been demonstrated to be feasible in the prehospital environment, whilst offering great insight into patient outcomes. Further studies should therefore look to incorporate HRV as part of their physiological monitoring, whilst also considering whether HRV control may offer therapeutic benefit.

Pharmacological interventions:

This systematic review identified three included studies that examined BP-lowering management among stroke patients in the prehospital setting. According to the RIGHT-2 trial, transdermal GTN therapy in stroke patients had no therapeutic effect, with regards to primary outcomes, as determined by mRS, and secondary outcomes included mortality rate and serious adverse events [13]. RIGHT-2 Investigators indicated that GTN produced a systolic reduction of 5.8 mmHg, a result that is lower than the reduction in the RIGHT study, which found a 21 mmHg SBP reduction [13]. Furthermore, Bath et al. [52] examined the effect of GTN among ICH patients as a part of the RIGHT-2 trial. It was reported that GTN administration was correlated with larger hematomas and adverse functional outcomes. These results indicate that the administration of GTN in ICH may enhance vasodilation or interfere with the hemostatic processes of ICH, which raises the question of whether alternative BP-lowering agents are effective. According to a post-hoc analysis of the Antihypertensive Treatment of Acute Cerebral Hemorrhage 2 trial (ATACH-2), intensive BP lowering in a hospital setting within two hours of onset was associated with improved functional outcomes and reduced ICH expansion. [53]. Hence, additional evidence on the impact of lowering BP in the ultra-early phase of stroke care is required, which motivated the new INTERACT4 (Intensive Ambulance-Delivered Blood Pressure Reduction in Hyper-Acute Stroke Trial) study [54]. 

In terms of neuroprotection strategies as part of acute stroke care, the FAST-MAG trial included 1700 participants and explored the benefit of administering magnesium sulfate as a neuroprotective agent for stroke patients in the prehospital setting within 2 h of onset. There was no noticeable difference in mortality and disability scores [44]. Recently, a systematic review of 4347 participants assessed the effects of initiating magnesium sulfate on stroke patients [55]. Following the initiation of magnesium sulfate treatment for stroke patients, neither functional outcomes nor the 90-day mortality rate were altered [55]. This suggests that magnesium sulfate administration in acute stroke may not be of benefit at present; however, the work in the FAST-MAG trial did demonstrate that ambulance-initiated treatment in stroke could be safe and feasible.

Hyperoxia during prehospital stroke care was examined through a retrospective analysis [45]. A total of 360 participants received oxygen therapy despite normal oxygen saturations [45], with hyperoxia being associated with significantly lower SBP, DBP, and MAP compared to the normoxic group for all stroke subtypes [45]. Interestingly, through analyzing the same study participants, Dylla et al. explored the effect of early oxygen therapy on stroke outcome [56]. It was demonstrated that neurological outcomes (defined as mRS and ambulatory status) did not differ among the three groups (hyperoxia, hypoxia and normoxia) [56]. Additionally, a retrospective analysis examined the reliability of oxygen administration to suspected stroke subjects in the ED and out-of-hospital setting and explored adverse events among these patients [57]. The patients were categorized based on the amount of oxygen administered in the ED: none, low-flow (2–4 L/min), and high-flow (10–15 L/min). In comparison with the low-flow and no-oxygen groups, the high-flow group had a lower occurrence of adverse events (defined as death rate, neurological deterioration status, or occurrence of ischemic event) [57]. Thus, neither of the retrospective studies reported any neurologic harm associated with oxygen administration. These studies demonstrated the potential for investigating oxygen administration as a promising and safe intervention in acute stroke care during the early stages. Capnography is a valuable adjunct to other patient monitoring in brain-at-risk states. With regards to the effect of measuring EtCO_2_ in acute stroke care, this review identified one study investigating its use in prehospital stroke patients [39]. Among the ICH and AIS groups, there was no significant difference in EtCO_2_ levels and survival rate [39]. Furthermore, 20 studies were included in a systematic review that examined the EtCO_2_ level in stroke patients in hospital. The study found that stroke patients are more likely to be hypocapnic [11]. In the early stages of acute stroke care, particularly in prehospital settings, this parameter has not received sufficient investigation yet, particularly given the vasoactive properties of carbon dioxide. Therefore, future prospective studies should consider the prehospital observation of EtCO_2_ and investigate whether changing variations alter patient outcomes. This is particularly relevant, as any observed differences could lead to investigation of the effect of simple maneuvers (such as hyper- or hypo-ventilation).

### 4.1. Strengths and Limitations

To our knowledge, we believe this is the first systematic review examining the presence of physiological variability and optimal monitoring strategies for these parameters in the prehospital stroke care setting. 

Our systematic review approach aligned with PRISMA guidelines. Nevertheless, several limitations of this systematic review should be addressed as regards the included studies, in particular, the degree of methodological heterogeneity, the lack of control populations in some studies, and small study sizes. Therefore, quantitative analyses were not performed. The NOS has been selected for a variety of studies because it is most suitable. However, it may not be the optimal choice for every study type.

### 4.2. Future Work

Considering the studies included, and in terms of the association between BPV and poor outcomes among stroke patients, future work is required to investigate the possibility of using BPV as an adjustable therapeutic target in acute stroke care. There are currently no benefits to be found in terms of mortality rates or functional outcomes for BP-lowering interventions in the prehospital setting. Therefore, additional information is needed on the influence of BP lowering on stroke patients. Additionally, HRV measurements revealed promising information that needs to be further investigated in future studies alongside predictive models for stroke outcome. 

## 5. Conclusions

In summary, this systematic review identified 19 records exploring the use of physiological parameter measurements for stroke patients in the prehospital environment. Key areas of interest highlighted in this review are BPV and HRV, as they may be associated with patient outcomes, however mixed results have been demonstrated on the effect of interventions relating to these, such as BP reduction. This review has also identified a lack of continuous physiological measurements prehospitally; as variability is more significant than isolated values, advancements in assessing these are a key interest for future work. Finally, this review has highlighted gaps in the current prehospital literature, through a lack of studies investigating key physiological parameters (such as EtCO_2_) and through comparison with recent comprehensive inpatient studies.

## Figures and Tables

**Figure 1 healthcare-12-00835-f001:**
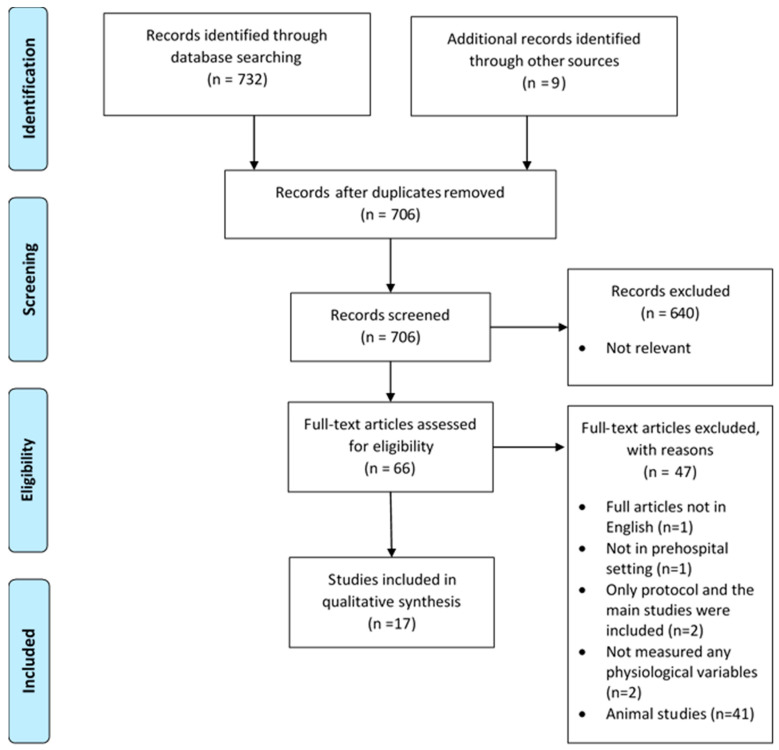
PRISMA flow diagram [23].

**Figure 2 healthcare-12-00835-f002:**
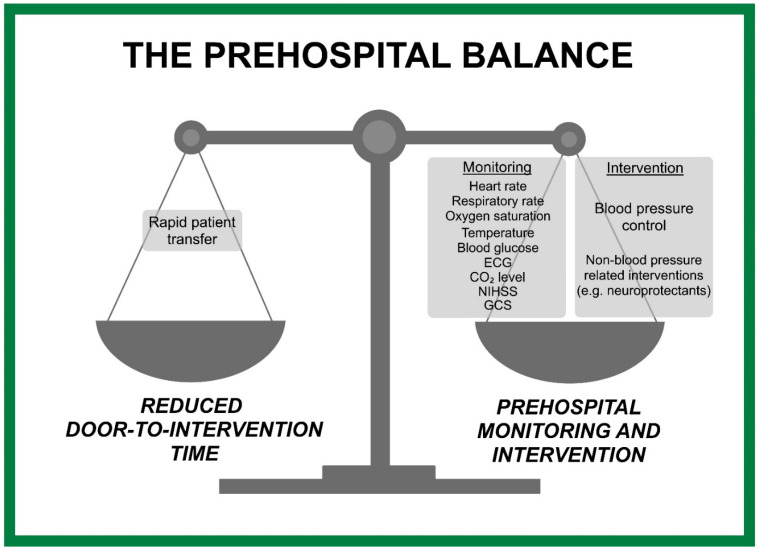
Balancing monitoring, intervention, and minimizing door-to-intervention time.

## Data Availability

Material and data included in manuscript and Appendix A.

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
