# Peer review of "Physiological Variability during Prehospital Stroke Care: Which Monitoring and Interventions Are Used?"

_healthcare, 2024, doi:10.3390/healthcare12080835_

Round 1

Reviewer 1 Report

Comments and Suggestions for Authors

This systematic review provides a comprehensive overview of physiological parameter monitoring and intervention within the prehospital care setting. 

Especially with regard to the introduction and the presentation of the results, I believe the manuscript would benefit from a revision.

*The authors mention the MSU concept in the introduction, and briefly under item 3.2.2. pharmacological therapy intervention 'Primary targets of therapy included BP reduction, neuroprotection, and mobile stroke unit. It is unclear and confusing what the authors mean by this. MSU is not a 'target of therapy', neither were MSU related papers included in the studies included in the qualitative synthesis.

*The rationale of the SR could be described more clearly. Statements such as "Monitoring of patient physiological parameters has been utilized in recent stroke advancements (such as MT) and has been described in recent literature" are not actually contributive and could be rephrased. 

*Please also consider summarizing the main findings in a table.

*As a minor comment, the manuscript should be revised for interpunction and typographical errors (eg. Suppl. mat Fig 1 "Comaparability")

Reviewer 2 Report

Comments and Suggestions for Authors

Dear authors,

thank you for giving me the opportunity to review your scientific article. 

Here are some suggestions for improvement:

I believe that abbreviations should not appear in the abstract.

It is surprising that only two reviewers assess the studies. How do an even number of reviewers reach a consensus?

The image relating to article selection is of poor quality and cannot be read.

Having heterogeneous studies is not a problem for a systematic review, only for a meta-analysis.

The topic is not very novel and too much emphasis is placed on blood pressure numbers, without going into other signs and symptoms.

As for the results found on drugs, there is a lack of further analysis of them.

It is not discussed whether, on the basis of the results, more time is needed for pre-hospital stabilisation of the patient or speed of arrival at a useful centre. This point is fundamental to this review.

In general, I think that the authors justify themselves too much on the impossibility of performing a meta-analysis and the conclusions reached do not meet the objectives of the study.

I hope this analysis will help them to improve the article.

The reviewer.

Reviewer 3 Report

Comments and Suggestions for Authors

thank you for the opportunity to review your manuscript

abstract

The abstract is far too long. it needs to be more concise (about half of what it currently is). It should also include some of the results from the review. 

Introduction. 

Line 61 - this is the basis of your systematic review - you need to provide at least a paragraph on why this systematic review is important. What associations have been suggested in the past?

Line 68 - this sentence does not make sense

Materials and Methods

No issues, have followed protocol

Results

Very comprehensive and well written - a lot of results would fit better within the discussion, especially when you start to link different studies together to describe solutions. 

Discussion

Slightly repetitive from the results. 

Conclusion

Reasonable conclusion and justification. 

Round 2

Reviewer 2 Report

Comments and Suggestions for Authors

Dear authors,

I consider that point five still needs to be improved.

Greetings.

The reviewer.
